# Inorganic Arsenite [As (III)] Represses Human Renal Progenitor Cell Characteristics and Induces Neoplastic-like Transformation

**DOI:** 10.3390/cells14120877

**Published:** 2025-06-10

**Authors:** Md Ehsanul Haque, Swojani Shrestha, Donald A. Sens, Scott H. Garrett

**Affiliations:** Department of Pathology, School of Medicine and Health Sciences, University of North Dakota, Grand Forks, ND 58202, USA; mdehsanul.haque@und.edu (M.E.H.); swojani.shrestha@ndus.edu (S.S.); donald.sens@und.edu (D.A.S.)

**Keywords:** arsenite, HRTPT, kidney progenitor, RCC, CD44

## Abstract

Arsenic, in the form of inorganic arsenite, is toxic to the kidney and can cause acute kidney injury, manifesting as destruction of proximal tubule cells. Nephron repair is possible through the proliferation of resident tubular progenitor cells expressing CD133 and CD24 surface markers. We simulated regenerative repair in the continued presence of i-As (III) using a cell culture model of a renal progenitor cell line expressing CD133 (*PROM1*) and CD24. Continued exposure and subculturing of progenitor cells to i-As (III) led to a reduction in the expression of *PROM1* and CD24, as well as a decrease in the ability to differentiate into tubule-like structures. Cessation of i-As (III) and recovery for up to three passages resulted in continued repression of *PROM1* and reduced ability to differentiate. Chronically exposed cells exhibited an ability to form colonies in soft agar, suggesting neoplastic transformation. Chronically exposed cells also exhibited an induction of CD44, a cell surface marker commonly found in renal cell carcinoma, as well as in tubular repair in chronic renal injury such as chronic kidney disease. These results demonstrate potential adverse outcomes of renal progenitor cells chronically exposed to a nephrotoxicant, as well as in environmental exposure to arsenic.

## 1. Introduction

Inorganic arsenic is a toxic substance that is distributed throughout the environment and has been implicated in numerous disease states; e.g., it has been shown to cause overt organ toxicity, especially in the liver and kidney; to exacerbate cardiovascular disease, neurological deficits, peripheral neuropathy, and diabetes; and to cause bladder, skin, lung, and other cancers [1]. While most attention is focused on i-As (III) exposure and cancer in many organ systems, an exception may be the kidney. In the United States, high levels of arsenic have been found in drinking water wells in more than 25 states, potentially exposing 2.1 million people to drinking water that is high in arsenic [2]. This number does not include other sources of arsenic exposure, such as food, soil, or drinking water with levels below those thought to be safe for human consumption. Globally, possibly the worst case ever of arsenic poisoning occurred in Bangladesh, where over 100 million people were poisoned by arsenic in groundwater supplies [3]. While renal cancer may be a concern, susceptibility of the kidney to i-As (III) may be more associated with alterations in renal function with progression to chronic kidney disease (CKD) [4,5,6,7,8,9]. Human bladder cancer is strongly associated with exposure to i-As (III) present in drinking water, and, since the kidney is intimately involved in the excretion of the metalloid upstream to the bladder, the cellular components of the kidney are exposed to at least the same level of i-As (III) as the bladder [10]. As reviewed by Kovesdy (2022), the number of patients affected by CKD is estimated at 843.6 million individuals worldwide as of 2017, and CKD has emerged as a leading cause of worldwide mortality [11]. Importantly, the kidney has the ability to repair and regenerate following nephron damage. Enhancing renal repair and regeneration is viewed as an important strategy in mediating the progression of individuals from acute tubular necrosis (ATN) to CKD. There is no information on how the process of renal repair and regeneration is influenced by exposure to i-As (III). Repair and regeneration of damaged tubular epithelium is presumed to be mediated by a sparse and widely dispersed population of renal progenitor cells within the kidney that co-express PROM 1 and CD24 [12,13,14,15,16,17]. Most studies assume that insults and damage only occur to mature and differentiated, functioning cells and not to regenerating progenitor cells. Understanding how progenitor cells themselves respond to injurious agents such as i-As (III) may be central in efforts to expand repair and regeneration of renal injury before progression to CKD.

A previous study from this laboratory described the isolation of an immortalized culture of human renal progenitor cells (HRTPTs) based on their co-expression of CD133 and CD24. The HRTPT cells also displayed the properties of growth and differentiation expected from in vivo and in vitro studies of CD133/CD24 human progenitor cells [18,19]. The HRTPT cells were used to demonstrate that exposure to i-As (III) caused the cells to shift from an epithelial morphology to a mesenchymal morphology. Genomic analysis over 10 serial passages (15 population doublings) confirmed a change in gene expression associated with a shift to a mesenchymal genotype. Interestingly, removal of i-As (III) at passage 10 allowed the cells to revert to an epithelial morphology and genotype like that of the control cells not exposed to i-As (III) [20]. This study also formed the foundation to undertake a longer-term study on the effects of i-As (III) exposure on the HRTPT cells. There is limited information regarding the lifespan of renal progenitor cells within the adult kidney [21]. The previous data demonstrated that the i-As (III)-exposed HRTPT cells at P_10_ still express PROM1 and CD24, albeit at a lower level, and can revert to the unexposed state when i-As (III) exposure is stopped. The goal of the study was to assess whether the cells would maintain or increase the repression of PROM1 and CD24 upon extended i-As (III) exposure and whether this could result in conversion to the loss of progenitor properties of the HRTPT cells with progression to those displaying characteristics of malignant transformation.

## 2. Materials and Methods

### 2.1. Cell Culture and Reagents

American Type Culture Collection (ATCC) provided parental RPTEC-TERT cells, and the HRTPT cell line was established as described previously [19,22]. A 1:1 mixture of DMEM/F12 serum-free media supplemented with selenium (5 ng/mL), insulin (5 μg/mL), transferrin (5 μg/mL), hydrocortisone (36 ng/mL), triiodothyronine (4 pg/mL), and epidermal growth factor (10 ng/mL) was used (Gibco, Waltham, MA, USA). The HRTPT cells were grown up to confluence and immediately incubated with 4.5 µM of i-As (III) (sodium arsenite) for 24 h (termed as “P_0_”) and then subcultured at a 1:3 ratio in the continued presence of i-As (III) until confluent. This process was repeated for 20 serial passages (P_19_). The passages were examined microscopically and cryopreserved, and harvested cells were preserved for RT-qPCR and protein analysis.

### 2.2. RNA Extraction and RT-qPCR

In order to determine mRNA expression levels of individual genes, RT qPCR was used as described previously [19,22,23]. RNA cell pellets were made by harvesting confluent cell cultures and freezing them under liquid nitrogen. Cell lysing was performed with 350 µL of RLT^®^ buffer (Qiagen, Hilden, Germany); they were dissociated using Qi-As (III) shredder tubes (Qiagen) for 2 min at 12,500 rpm. RNA extraction was carried out using QIAGEN’s RNeasy Mini Plus Kit (#74034) and QIAcube instrument (Hilden, Germany) according to the manufacturer’s protocols. Quantification of RNA was performed using a NanoDrop spectrophotometer (Thermo Fisher Scientific, Waltham, MA, USA). cDNA was generated from total RNA by using LunaScript ^®^ RT SuperMix Kit (New England Biolabs #E3010L, Ipswich, MA, USA) following the manufacturer’s protocols. cDNA was diluted to a final concentration of 20 ng/L with nuclease-free water. The BioRad CFX96 Touch Real-Time PCR Detection System) (Hercules, CA, USA) and the Luna^®^ Universal qPCR Master Mix (New England Biolabs #M3003E) were used to perform the qPCR using only two microliters of cDNA (20 ng). The protocols for the qPCR cycle included one 2 min cycle at 95 °C, forty 5 s cycles at 95 °C, and a 30 s annealing step at 60 °C temperature. The 18S ribosomal RNA (rRNA) is a structural RNA used as housekeeping gene, and the expression levels were determined by the threshold cycle (Ct) values through the 2-∆∆Ct method.

### 2.3. Western Blot

Western blot analysis was used to evaluate protein expression using standard methods, which has been previously published [19,22,23,24]. Following cell lysing protocols with an ice-cold RIPA buffer with equal amounts of protease inhibitors, sodium orthovanadate (Santa Cruz), and phenylmethylsulfonyl fluoride (PMSF), the cell pellets (HRTPTs) were shaken for 15 min on ice. After being sheared twice for approximately 15 s with a sonicator while being kept on ice, the extracts were centrifuged for 10 min at 4 °C at 13,000× *g*. After transferring the supernatants to brand-new, cool micro centrifuge tubes, the BCA assay (ThermoScientific) was used to quantify the proteins. Each sample was diluted to 50 μg using RIPA, combined with Laemmli buffer (Bio-Rad, Maharashtra, India), and boiled for five minutes at 95 °C after being quantified, if necessary. The materials were loaded onto a TGX Any Kd SDS polyacrylamide gel (Bio-Rad Laboratories) after being rapidly centrifuged for 10 s. They were then moved to a nitrocellulose membrane (Amersham Biosciences, Piscataway, NJ, USA). After being rinsed for five minutes in TBST, the blots were blocked for one hour at room temperature the same day in Tris-buffered saline (TBS) with 0.1% Tween-20 (TBS-T) and 5% [*w*/*v*] nonfat dry milk. After blocking, the membranes were probed with the matching primary antibody for a whole night at 4 °C in a shaker after three 15 min TBS-T washes. The appropriate dilution factor was used to dilute the primary antibodies, which were prepared using either 3% BSA or 5% non-fat milk in TBS-T.

### 2.4. Capillary Electrophoresis Western

Mehus et al. 2023 described a simple Western blot method for measuring and analyzing protein expression [23]. Bicinchoninic Acid (BCA) protein assay kits (Thermo-Scientific Pierce, Waltham, MA, USA) were used to determine protein concentration in the samples. Protein lysates were diluted and mixed with 5× fluorescent master mix (ProteinSimple, San Jose, CA, USA) that contained dithiothreitol, fluorescent standards, and a system control protein (26 kDa). The protein mixture was then heat-denaturated (95 °C for 5 min). In order to detect target proteins, protein lysate was separated and immunodetected using a capillary-based Jess Simple Western instrument (ProteinSimple, San Jose, CA, USA) according to the manufacturer’s protocol. The control protein (26 kDa) served as an internal control and was also used to normalize protein expression. For each sample, a quantity of 4 µL (0.5 µg/µL) was loaded for target protein expression. Following a calibrated dilution, the concentrations of the protein lysate and antibodies were used in the electrophoresis.

### 2.5. Immunofluorescence

Cells were grown on number 1.5 (0.17 mm thickness) coverslips to confluency and fixed with 3.7% paraformaldehyde (pre-warmed at 37 °C) in PBS (500 µL/each well) for 15 min, followed by three PBS washes each for 5 min [25,26]. Next, permeabilization was carried out using 500 µL of 0.1% Triton-X 100 in 1× PBS for 10 min. The coverslips were washed three times using PBS for 5 min followed by blocking with 1% BSA in 1× PBS for 30 min, rinsed quickly with 2 PBS washes with 1× PBS, rinsed three times for 5 min, and incubated with primary antibody (1% BSA in 1× PBS, 66 µL of antibody solution/per cover slip, vortex to mix and spin briefly) for 45 min at room temperature. The primary antibodies were diluted in PBS, and an appropriate dilution factor was utilized. The cells were washed three times with PBS for 3 min each. Secondary antibody (60 µL) was added to the coverslip and incubated for 45 min at room temperature. The primary antibody was detected by incubating cells with Alexa-Fluor 488 secondary antibody for 30 min at 37 °C. The coverslips were washed three times with PBS for 3 min and mounted upside down with Prolong Diamond Antifade Mountant with DAPI (Life Technologies, Carlsbad, CA, USA). The stained cells were observed and imaged using an Olympus FV 3000 confocal microscope. Two coverslips per sample were set up, and a minimum of 5 fields per coverslip were examined.

### 2.6. Sphere Formation Assay

Confluent culture of iAs (III)-exposed HRTPT (P_15_) cells and unexposed-HRTPT cells was detached with the BD Biosciences accutase enzyme (Cat # 561527, 2 mL for each T25 flask) for 15 min and centrifuged with 4 mL of PBS (2000 rpm, 3 min) [19,22,27,28]. Subsequently, PBS was removed, cell pellets were resuspended in 1 mL media, and the cell count was performed by an automated cell counter (Countess3, Invitrogen) (trypan blue 50 μL + 50 μL of cell suspension media); 5000 cells were seeded into two six-well Corning™ ultra-low attachment flask (#03723038, Thermo Fisher Scientific) plates supplemented with 2 mL of serum-free growth medium. Cells were allowed to grow undisturbed for ten days at 37 °C in a 5% CO_2_ incubator. The mechanical counter was used to count the number of spheres under a microscope for each condition. Three biological replicates were performed for each condition. To determine the statistical significance, a one-way ANOVA was used on GraphPad.

### 2.7. Matrigel™ Tubular Differentiation

A 48-well plate was coated with 250 µL of Okamatrix Matrigel (#354230, OkaSciences, Kelowna, BC, Canada) before solidifying for 30 min [19,22,27,28]. Each condition was seeded with 5000 cells per well, with three replicates of iAs (III)-exposed HRTPT (P15) cells and unexposed HRTPT cells. In the following steps, 500 µL of serum-free culture medium was supplemented with 4.5 µM iAs (III), and control cells without iAs (III) were added on top of each condition. Cells were allowed to attach for 72 h at 37 °C in a 5% CO_2_ incubator. The medium was refreshed every 48 h as the cells grew. After seven days, 14 days, and 21 days, tubular formation was monitored using light microscopy.

### 2.8. Soft Agar Assay

This experiment was performed following previously described protocols [29,30,31,32]. The bottom layer consisted of 0.5% agar in growth medium, and the top layer consisted of 0.4% agar in growth medium. Cells were suspended in the 0.4% growth medium at 500 cells per dish prior to pouring the top layer. Petri dishes were placed in an incubator on a tray alongside an autoclaved water beaker. Cell clumps or colonies (8–10 cell aggregates were defined as a cluster) were counted using a dissecting microscope after four weeks. Depending on the rate of cell cluster development, the plates were incubated for 21 to 30 days at 37 °C in a humidified incubator.

### 2.9. In Vitro Cell Differentiation Assay

DMEM/F12 medium was used in the culture of non-exposed controls, P_3_-As (III) removal, and P_19_-As 4.5 µM-HRTPT cells to confluence, followed by conditions favorable for tubulogenic, osteogenic, adipogenic, and neurogenic cell induction, as previously described by Lazzeri et al. [27,33]. The tubulogenic differentiation was achieved by culturing non-exposed controls, P_3_-As (III) removal, and P_19_-As 4.5 µM-HRTPT cells for 2–3 weeks in REBM medium (Lonza, Basel, Switzerland) supplemented with 50 ng/mL hepatocyte growth factor (HGF) (Peprotech). To achieve osteogenic induction, non-exposed controls, P_3_-As (III) removal, and P_19_-As 4.5 µM-HRTPT cells were cultured in α-MEM and 10% horse serum that contained 100 nM of dexamethasone, 50 µM of ascorbic acid (Sigma-Aldrich), and 2 mM of β-glycero-phosphate (Santa Cruz Biotechnology, Santa Cruz, CA, USA). The medium was changed twice a week for two to three weeks. In order to promote neurogenic differentiation, non-exposed controls, P_3_-As (III) removal cells, and P_19_-As 4.5 µM-HRTPT cells were grown in DMEM, hg, and 10% FBS for 24 h and replaced with DMEM, hg, 10% FBS containing 20% B27 (Gibco), 10 ng/mL EGF, and 20 ng/mL basic fibroblast growth factor (Peprotech). The medium was changed to serum-free DMEM containing 200 mM of indomethacin, 5 µg/mL of insulin, and 0.5 mM of 1-methyl-3-isobutylxanthine five days later. Adipogenic induction was achieved by growing non-exposed control cells, P_3_-As (III) removal cells, and P_19_-As 4.5 µM-HRTPT cells in DMEM, high glucose (Gibco, Ireland), containing 10% FBS, 1 mg/mL dexamethasone, 0.5 mg/mL 1-methyl-3-isobutylxanthine, 10 µg/mL insulin, and 100 µM of indomethacin (Sigma-Aldrich). The medium was changed after 72 h, and the cells were maintained in DMEM hg, 10% FBS, and 10 µg/mL insulin until confluence growth.

### 2.10. Strategies for Arsenite Removal from Freeze-Down Chronic Arsenite-Exposed Cells

Freeze-down chronic arsenite-exposed cells were taken out of the refrigerator at -80 °C and thawed in a water bath at 42 °C [20]. Cell pellets with DMSO were placed into 15 mL Falcon tubes, dissolved in 9 mL of regular DF12 media, and then centrifuged for 3 min. Supernatant was removed with a suction pipe, and pellets were dissolved in regular DF12 media and 4.5 µM of i-As (III). When cells were confluent, they were washed twice with PBS buffer and fed with regular DF12 media for 24 h (P_0_-Arsenite removal). The next day, cells were harvested for RNA, as well as protein, and other flasks were split into P_1_ (1:3 ratio) in i-As (III)-free growth media; they continued to grow in i-As (III)-free media for the next P_3_ passage.

### 2.11. Tumor Formation in Immunocompromised Mice

The ability of the iAs-exposed cells to form tumors was tested by the injection of 1 × 10^6^ cells at passage 16 in the dorsal thoracic midline of athymic nude mice (NCRnu/nu) as described previously [34]. The mice were observed every two weeks for 12 weeks for any sign of tumor formation (raised area at the injection site). Following euthanasia, the injection site of all 5 mice was dissected, and the area was observed for any evidence of tumor growth. This study was approved by the UND IACUC #1911-1C.

### 2.12. Statistical Analysis

The experiments were carried out in triplicate, and the data were analyzed using one-way ANOVA (non-parametric) with Tukey post hoc testing using GraphPad PRISM 10. Gene expression was normalized to the 18S housekeeping gene. The measurements were performed in triplicate for gene data. The reported values are mean ± SEM. A *t*-test was performed for mRNA and protein, respectively. And asterisks indicate significant differences from the control (* *p*< 0.05, ** *p* < 0.01, *** *p* < 0.001, **** *p* < 0.0001).

## 3. Results

### 3.1. Growth, Morphology, and PROM1 and CD24 Expression of HRTPT Cells Exposed to 4.5 µM of i-As (III) for 30 Population Doublings

The exposure of the HRTPT cells to 4.5 µM of i-As (III) followed the identical protocol described previously, except that the i-As (III) exposure was extended from P_8_ to P_19_ [20]. Table 1 shows that during the initial passages (P_1_ to P_9_), the cells struggled to reach confluence in the continued presence of the metalloid, with five of the passages requiring more than one month to fill the flask, and P_3_ requiring three months. In later passages (P_10_ to P_19_), the cells were able to grow and divide during continued exposure to 4.5 µM of i-As (III) at a much greater rate, requiring only between one and two weeks per passage.

The light-level morphology of the i-As (III)- exposed HRTPT cells maintained a mesenchymal-like morphology at P_8_ through P_19_ when compared to the HRTPT control (Figure 1). The days to confluence following a 1:3 subculture of the i-As (III)-exposed cells compared to control (7 days) showed that the time to confluence for the i-As (III)-exposed cells was increased at P_8_ (65 Days) and P_12_ (12 days) but was similar to that of the control cells at P_18_ and P_19_. This result shows that at late passages, the i-As (III)-exposed HRTPT cells increase their growth in a manner that is similar to the control cells. The cells at P_19_ showed no evidence of multilayer growth or of the formation of raised foci of cells above the monolayer.

An analysis of PROM1 expression showed that the i-As (III)-exposed cells had a significantly reduced expression of PROM1 mRNA and low levels of PROM1 protein (Figure 2; A and B). A similar analysis for CD24 showed a reduced expression on mRNA and protein at passages 8 and 15, with a strong rebound in CD24 mRNA and protein at passage 19 (Figure 3).

### 3.2. Progenitor Properties of HRTPT Cells Exposed to 4.5 µM of i-As (III) for 15 Serial Passages/Population Doublings

The i-As (III)-exposed HRTPT cells at P_15_ retained the ability to form spheroids when subcultured into ultra-low attachment culture dishes (Figure 4). The number of spheroids generated by the i-As (III)-exposed cells was significantly increased compared to control cells not treated with i-As (III). This suggests that the i-As (III)-exposed cells at P_15_ retain the property of self-renewal at a significantly reduced expression of PROM1 protein. A further test of progenitor properties involved the ability of the i-As (III)-exposed cells to form tubule-like structures when grown on the surface of Matrigel (Figure 5). Compared to control cells able to form tubule-like structures, the i-As (III) cells at P_15_ showed no evidence of the ability to form these structures (Figure 5). The ability of the i-As (III)-exposed cells at P_15_ to undergo differentiation was tested using gene expression markers commonly used to provide evidence for tubulogenic, adipogenic, osteogenic, and neurogenic differentiation in HRTPT cells [18]. CD133^+/^CD24^+^ cells, isolated from human kidney, have been demonstrated to transdifferentiate into these cell types [35]. The results showed that the markers for tubulogenic differentiation (AQP1) and adipogenic differentiation (ADIPO) were significantly elevated, and the marker for osteogenic differentiation (RUNX2) significantly decreased compared to control cells (Figure 6). The markers for neurogenic differentiation (ENO2 and NES) were close to the value of the control cells but did not reach statistical significance. Except for osteogenic differentiation, the i-As (III)-exposed cells at P_15_ retained their ability to differentiate when placed in an appropriately defined growth formulation.

### 3.3. Epithelial–Mesenchymal Transition (EMT) in HRTPT Cells Exposed to i-As (III) for 19 Passages (30 Population Doublings)

The previous study exposing HRTPT cells to i-As (III) at shorter serial passages suggested a shift to mesenchymal morphology and a gene expression profile suggesting the development of EMT [20]. An analysis of the HRTPT cells exposed to i-As (III) for 20 passages was assessed for E-cadherin (CDH1), N-cadherin (CDH2), and vimentin (VIM) to determine if the shift to EMT had undergone progression. The results show a marked reduction of E-cadherin and a marked increase in N-cadherin and vimentin mRNA expression, further confirming the development of EMT (Figure 7). The expression of CDH1 mRNA was similar to the control through P_11_ and then decreased sequentially starting at P_12_ through P_20_ (Figure 3A). The CDH1 protein was also reduced but lagged behind CDH1 mRNA, presumably due to the time of degradation of the CDH1 protein (Figure 7B,C). The expression of CDH2 mRNA remained at the low expression levels found in control cells through P_13_ and then increased significantly by P_20_. CDH2 protein was similarly increased, again with a lag time between mRNA and protein expression (Figure 7D,E). The pattern of expression of VIM mRNA and protein was similar to that of CDH2 (Figure 7H,I). Confocal images of CDH1, CDH2, and VIM were obtained at P15 and show equivalent localization for CDH1 and CDH2 at this cross-over point of protein expression, with VIM showing an increase in protein expression (Figure 7F,G,J). The results show that EMT progressed as the time of i-As (III) exposure increased for the HRTPT cells.

### 3.4. As^3+^ Anchorage-Independent Growth and CD44 Expression of HRTPT Cells Exposed to i-As (III) for 19 Passages (30 Population Doublings)

The soft agar colony formation assay is an established method to evaluate anchorage-independent growth of cells (independent of a solid surface), and it is an established hallmark of carcinogenesis and tumorigenic potential [29,30]. The results of this study showed that the i-As (III)-exposed HRTPT cells at P_19_ were able to form colonies in soft agar (Figure 8B). The control cells showed no evidence of the formation of colonies in soft agar (Figure 8A).

An attempt was made to assess tumor formation in vivo using cells passaged in 4.5 μM of i-As (III) from our previous study [34]. The examination of the five mice injected with iAs-exposed cells at P_16_ showed no visible evidence of tumor formation, nor was any visible evidence observed following examination of the tissue at the injection site.

The CD44 gene was also examined for its expression in the i-As (III)-exposed HRTPT cells. This gene was chosen because it is a known cancer stem cell marker with prognostic significance for patients with clear cell renal carcinoma [36]. The results show that control cells have a low expression of CD44 compared to i-As (III)-exposed cells at P_3_ through P_20_, with P_20_ cells displaying a 6-fold increase compared to the control (Figure 9A). The level of CD44 protein expression increased 150-fold over the control at P_15_. Confocal analysis showed a marked increase in the cell surface expression of CD44 for the i-As (III)-exposed cells at P_15_ compared to the control.

### 3.5. Effect of iAs (III) Removal from HRTPT Cells Exposed to iAs (III) for 19 Passages

The effect of removing i-As (III) from the growth media was tested for 3 passages on the HRTPT cells exposed to i-As (III) for 20 passages (23 passages of total cell growth). The light-level morphology of the cells was assessed for control cells, i-As (III)-treated cells at P_20_, and i-As (III)-treated cells following the removal of i-As (III) from the growth media for three serial passages (Figure 10). The results show that the i-As (III)-exposed cells at P_20_ and the three passages where i-As (III) was removed displayed a similar morphology. The control cells never exposed to i-As (III) had a more epithelial morphology and produced domes, a feature only occurring in epithelial cells (Figure 10A). The i-As (III)-exposed cells and those with i-As (III) removed displayed no domes and a more mesenchymal morphology. The three passages of the P_20_ cells where i-As (III) was removed retained the ability to form spheroids on low attachment flasks and to form colonies in soft agar and the inability to form tubule-like structures on the surface of Matrigel (Figure 11, Figure 12 and Figure 13). The cultures that were placed in i-As (III)-free media for three passages showed an increase in CDH1 mRNA compared to the control, but CDH1 protein had a similar expression compared to the control. The expression of CDH2 and CD44 mRNA and protein was significantly higher than the control for the P3 cells in the absence of i-As (III) (Figure 14). When compared to i-As (III)-exposed cells at P20, CDH1 mRNA and protein were elevated, and CDH2, CD44 mRNA, and protein remained elevated at levels similar to i-As (III)-exposed cells at P20 (Figure 14). The expression of CD133 mRNA and protein was reduced compared to control cells and similar to i-As (III)-exposed cells at P20. The expression of CD24 mRNA and protein at P3 was similar to that of the control and P20 i-As (III)-exposed cells (Figure 15).

## 4. Discussion

The present study shows that i-As (III) exposure can induce EMT in HRTPT cells that co-express PROM1 and CD24 and demonstrate the properties expected of a renal progenitor cell. The two most frequently studied markers of EMT are E-cadherin (CDH1) and N-cadherin. CDH1 is one of the main epithelial markers that is downregulated in the process of EMT. CDH2 is contradictory to CDH1, and its increased expression is a marker of ongoing EMT [37,38]. Cadherin switching to high N-cadherin and low E-cadherin expression is essential for EMT. VIM functions as a positive regulator of EMT, and upregulation is strongly related to the induction of EMT [39]. The expression of these markers in the HRTPT cells exposed to i-As (III) for 20 passages reinforces the notion that i-As (III) exposure caused the cells to undergo EMT. Three types of EMT have been defined; type 1 is involved in embryogenesis and organ development, type 2 is associated with tissue regeneration, wound healing, and organ fibrosis; and type 3 is implicated in cancer progression through the occurrence of genetic and epigenetic alterations promoting clonal outgrowth and the formation of localized tumors [40]. Within these definitions of EMT, there is ambiguity beyond just cadherin switching due to evidence of partial EMT in both type 2 and type 3 EMT, making it difficult to determine where the i-As (III)-exposed HRTPT cells fall within this process. Type 1 EMT can be eliminated from consideration, since it is involved in embryonic development. There appears to be considerable overlap between type 2 and 3, as many associated markers are present in both types. An example of this in the present study is the increased expression of CD44 over 30 population doublings of the i-As (III)-exposed cells. Increased expression of CD44 is both an independent risk factor for patient death in ccRCC and is also found in renal tubular cells undergoing repair and regeneration [36,41]. This overlap in expression was also shown for the i-As (III)-exposed HRTPT cells when gene expression correlated with cycle tubules in damaged kidneys and with gene expression in cancer cell lines. At least for cells in culture, the only definitive separation is if the i-As (III)-exposed cells form tumors upon xenotransplantation in immune-compromised mice. Since the HRTPT cells exposed to i-As (III) for 20 passages did not form tumors, the requirement for a fully developed EMT in type 3 EMT falls short. However, the HRTPT cells retained the ability for spheroid formation and growth in soft agar. Thus, the HRTPT cells exposed to i-As (III) for 20 serial passages fall somewhere between type 2 EMT involved in renal repair and type 3 EMT that is progressing to tumorigenicity. The fact that the iAs-exposed cells can be consistent with both type 2 and 3 EMT is best explained by the lifespan of the cells. While 20 passages can be looked upon as a long time in cell culture, it only represents 30 population doublings out of an accepted lifespan of at least 60 doublings for a normal cell [42]. At 30 population doublings, it is possible that alterations between type 2 and 3 EMT are highly similar. Multistep carcinogenesis is a long process, and only one clonal event leads to malignancy. The fate of cells that do not clonally evolve to malignancy in this process is largely unknown.

The finding that the results can encompass both type 2 and type 3 EMT is not mutually exclusive. Exposure to i-As (III) is implicated in both CKD and ccRCC [4,6,7,43,44]. However, what is important is that the i-As (III)-exposed cell at extended passages still maintained the expression of PROM1 and CD24 along with properties expected of a progenitor cell. Theoretically, this means that the HRTPT cells at extended i-As (III) exposure could progress to damaged cells associated with the development of CKD or ccRCC. Cells co-expressing PROM1 and CD24 have been isolated from ccRCC tumors as a small overall component of the tumor but with progenitor properties [45]. Their low abundance in ccRCC tumors would likely be below that detectable using immunohistochemical studies of formalin-fixed, paraffin-embedded tissue from patients treated for ccRCC. Like their normal adult renal progenitor counterparts (ARPCs), these RCC-derived cells (RDCs) displayed self-renewal ability, clonogenic multipotency, and stemness-related elements. Like the i-As (III)-exposed HRTPT cells, the PROM1-expressing RCC cells did not form tumors upon xenotransplantation, but when co-transplanted with RCC cells, they enhanced tumor engraftment, vascularization, and growth [46].

Implicating i-As (III) exposure with ARPCs and the progression to CKD does not have a convenient endpoint like cancer formation, since the involved cells are replaced by fibrotic tissue. It has been known for many years that the kidney is susceptible to acute exposure to i-As (III) [47]. The role of i-As (III) exposure in the development of CKD relies on epidemiological studies on human populations [6,48]. These two studies from Taiwan assessed low to moderate exposures to i-As (III). Other than the previous publication from this laboratory [20] and the current study, there are few, if any, long-term or acute studies on the effect of i-As (III) or other toxicants on human ARPCs. The immortalized HRTPT cell line was used to close this gap in the literature, as it can undergo extended serial passage, and the individual cells co-express PROM 1 and CD24 as determined by flow cytometry. These properties allowed the first in vitro examination of i-As (III) exposure to human ARPCs over an extended period in culture and clearly showed that i-As (III) can produce EMT of the ARPCs. The removal of i-As (III) after 20 serial passages provided evidence that the i-As (III)-induced changes remained for at least three passages in the absence of i-As (III). This included the large increase in the expression of CD44 that occurred upon extended passage of the i-As (III)-exposed cells. The i-As (III) was removed from the confluent cells 48 h prior to the initial subculture. This is in contrast with the earlier study where i-As (III) removal rapidly returned the HRTPT cells to the morphology and the gene expression pattern of the parent control cells [20]. This provides initial evidence that the long-term changes due to i-As (III) exposure might be long-lasting or permanent with the retention of progenitor properties.

CD44 fared prominently in the current study, having roles in long-term responses to injury and fibrosis and being a prognostic marker for various cancers, including ccRCC. CD44 is a cell surface transmembrane glycoprotein that functions in cell adhesion, migration, and lymphocyte homing. It is expressed on a variety of cell types, including granulocytes, lymphocytes, fibroblasts, and even in epithelial cells. Functional ligands include hyaluronic acid, osteopontin, collagens, and matrix metalloproteinases, and CD44 activation initiates a variety of cell signaling pathways through interactions with intracellular kinases. Many of these activities and pathways are also commonly exploited by tumor cells to influence tumor growth, metastasis, and resistance to therapy. Thus, CD44 is a common marker of various types of cancers, including renal cell carcinoma. Important for the current study is the role of CD44 in the process of fibrosis. Since CD44 has many interactions with the extracellular matrix, there are many potential ways that this surface protein can facilitate processes in fibrosis, such as the recruitment of fibroblasts, promoting the deposition of matrix, and the recruitment of inflammatory cells [49,50]. Recently, Matsushita et al. (2024) showed that CD44 promoted renal fibrosis by secreting fibronectin from renal tubular epithelial cells (TECs) during partial epithelial–mesenchymal transition in response to renal injury [41]. CD44 was expressed in atrophic, dilated, and hypertrophic TECs with fibrotic lesions during chronic injury. By laser microdissection, these TECs were collected and analyzed using microarrays. Gene ontology analysis suggested that these TECs had a mesenchymal phenotype, and pathway analysis identified CD44 as an upstream regulator of fibrosis-related genes, including fibronectin 1 (Fn1). This study shows that CD44 expression marks renal tubular epithelial cells undergoing maladaptive repair and promotes the development of renal fibrosis [41]. For roles in cancer, most solid malignancies contain cancer stem cells characterized by CD44. As a result of the interaction of hyaluronic acid with CD44, EGFR-mediated pathways are activated, leading to tumor cell growth, tumor cell migration, and chemotherapy resistance in solid cancers [51].

Other researchers observed that an active CD44/p-AKT/AKT signaling pathway promotes resistance to FGFR1 inhibition in squamous-cell lung cancer [52]. An in-depth study of the roles of CD44/CD24 and ALDH1 as cancer stem cell markers in tumorigenesis and metastasis [53,54] showed that the CD44 gene is a major molecular prognostic marker for cancer stem cells (CSCs) of many solid malignancies, including RCC, as well as one of the major components of surviving tubular cells in renal fibrosis [36,53,55,56]. The expression of CD44 in clear cell renal cell carcinoma (ccRCC) correlates with tumor grade and patient survival, influenced by the methylation of genes [57]. Furthermore, higher CD44 expression is associated with more aggressive behavior, tumor progression, and a worse prognosis for RCC [53,58,59]. The current study shows that CD44 was induced in the late phases of the exposure timeline and appears to have a dual role in promoting maladaptive and fibrotic-like repair, as well as in promoting transformation. It is important to note that, according to Figure 9D, CD44 was expressed in nearly all epithelial cells, but in Figure 12, only 5 to 10% of the cells were able to form colonies in soft agar. Thus, CD44 is likely to function in promoting fibrosis during chronic injury for the majority of cells, and for those cells that have undergone transformation, to potentially enhance their tumorigenicity and/or progression.

A major role of cell culture is to provide in vitro clues for subsequent examination in human disease, in this case CKD. The limitations of cell culture are well known and include the lack of interaction with the ECM, the immune system, the adjacent cell types, and the secreted factors from other organs and distant cells. Conversely, these disadvantages also provide an advantage for the testing of effector agents. The serum-free growth media allows testing without the interference from fetal calf serum. Cell culture provides an avenue to test the consequences of renal toxicants on progenitor cells. In addition, one limitation of the current study is that the analysis is performed on an entire culture of cells. This results in a limited ability to determine the heterogeneity of the cells, resulting in individual measurements being the average of all cells. Thus, there could exist interesting subpopulations of progenitor cells unable to be identified in the cell population. Similarly, distinct cell populations could emerge over time that have a proliferative advantage and become dominant as the number of passages increases. However, to date, there are very few studies using a system where a culture system is composed of only cells co-expressing CD24 and PROM1. One study that was identified used primary cultures of ARPCs in co-culture experiments to demonstrate ARPCs could mediate cisPt damage to a putative culture of human proximal tubule cells [60]. The finding that the reduction in toxicity was mediated by exosomes from the ARPC demonstrates a novel way that ARPC may alter adjacent cell behavior. Overall, primary and immortalized cultures of ARPC are needed to study how external agents might affect renal repair and regeneration.

Another limitation in this study was a result of our difficulty in finding a suitable control. We felt that passage-matched controls, in particular, were not proper controls, since the exposed cells took several weeks to several months to reach confluency during arsenite exposure. Without arsenite exposure, the cultures reach confluency in one week, whereas with the exposures, they require several weeks, and passage 8 required 65 days, as is now shown in Table 1. It is difficult to have a control sample to match for this. These cells struggle to gain confluency over a couple of months. Simply having the cells harvested with the equivalent number of doublings is not really a direct control. Cells seldom change in several passages, and the overwhelming issue with arsenic exposure was to overcome the toxicity to fill the flask. The passage-match for that culture is really quite divergent from the experimental group. We think that the earlier passages with arsenite exposure are the true control because these cells have gone through the struggle of reaching confluency with arsenite but have yet to undergo the molecular changes in the various genes we are measuring.

## Figures and Tables

**Figure 1 cells-14-00877-f001:**
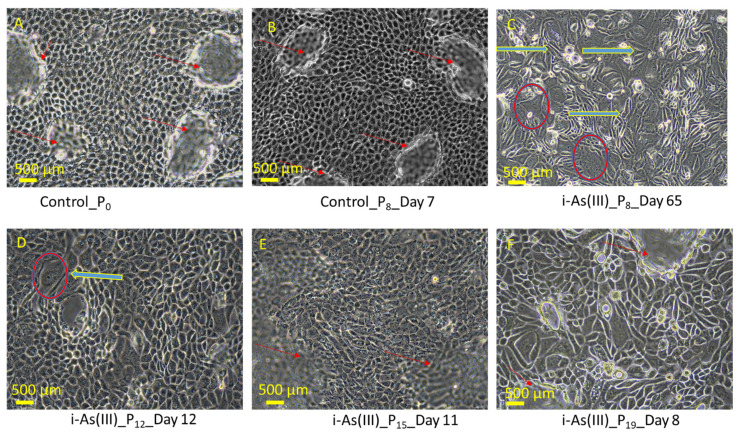
Light microscopic images of HRTPT cells exposed to 4.5 µM of arsenite show EMT-like morphological change at passage at P8; (**A)** (Control_P_0_) and (**B**) (Control_P_8__Day 7) are controls-unexposed cells; (**C**) (P_8-_i-As^3+^_Day 12), (**D**) (P_12-_i-As^3+^_Day 11), (**E**) (P_15-_i-As^3+^ _Day11), (**F**) (P_19_-i-As^3+^_Day8) cells exposed to 4.5 µM of arsenite passaged up to P_20_ (including P_0-_24 h of arsenite exposure) and grown in 1:1 DMEM/F12 media. Red arrows show dome-like structures, blue arrows show fibroblast-like structures, and red circles show myofibroblast-like cells, scale bar = 500 µm and magnification × 20.

**Figure 2 cells-14-00877-f002:**
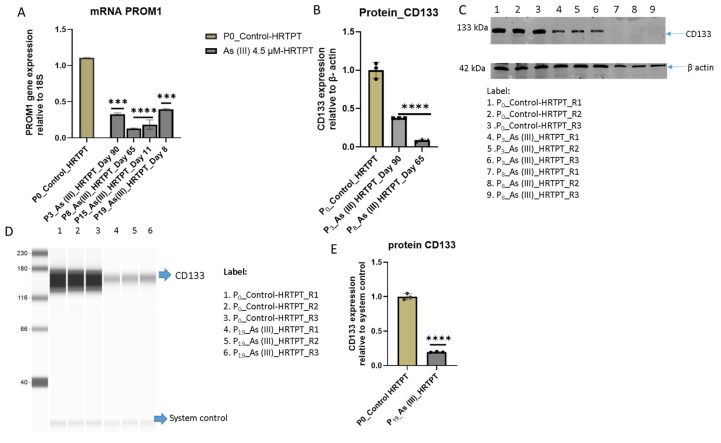
mRNA and protein level of PROM1 in HRTPT cell lines treated with 4.5 µM of i-As up to P19 passages. (**A**) mRNA PROM1-RT-qPCR analysis, (**B**) protein CD133 analysis (quantification from image **C**), (**C**) Western blot, (**D**) capillary Western/Jess of CD133, (**E**) protein CD133 analysis (quantification from image **D**); the expression of the PROM1 gene was normalized to the 18S housekeeping gene. While the expression of the CD133 protein was normalized to β-actin in the Western blot and to the system control in the capillary Western. Protein expression in (**D**) was also normalized to the Jess system control. The measurements were performed in triplicate for gene and protein data. The reported values are mean ± SEM. A one-way ANOVA was performed, and **** and *** indicate significant differences in gene/protein expression level compared to the control (0.0 µM arsenite concentration at *p*-value of ≤0.0001; ≤0.001, respectively).

**Figure 3 cells-14-00877-f003:**
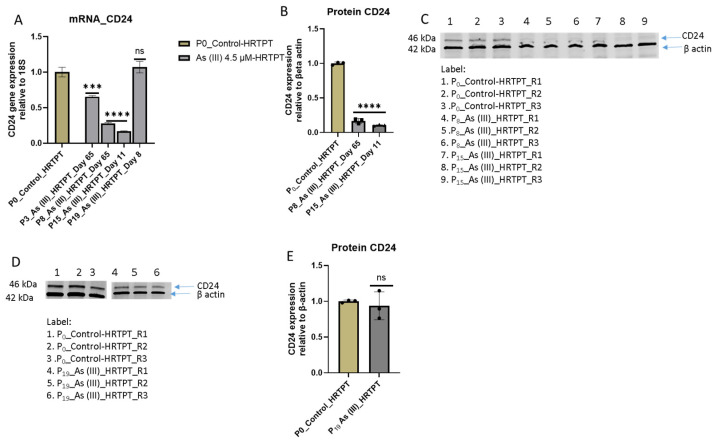
mRNA and protein level of CD24 in HRTPT cell lines treated with 4.5 µM of i-As (III) up to P_19_ passages. (**A**) mRNA_CD24-RT-qPCR analysis, (**B**,**E**) protein analysis of CD24, (**C**,**D**) Western blot of CD24 The expression of the CD24 gene and the protein were normalized to the 18S housekeeping gene and β-actin, respectively. The measurements were performed in triplicate for gene and protein data. The reported values are mean ± SEM. A one-way ANOVA was performed, and **** and *** indicate significant differences in gene/protein expression level compared to the control (0.0 µM arsenite concentration at *p*-value of ≤0.0001; ≤0.001, respectively). “ns” indicates non-significant.

**Figure 4 cells-14-00877-f004:**
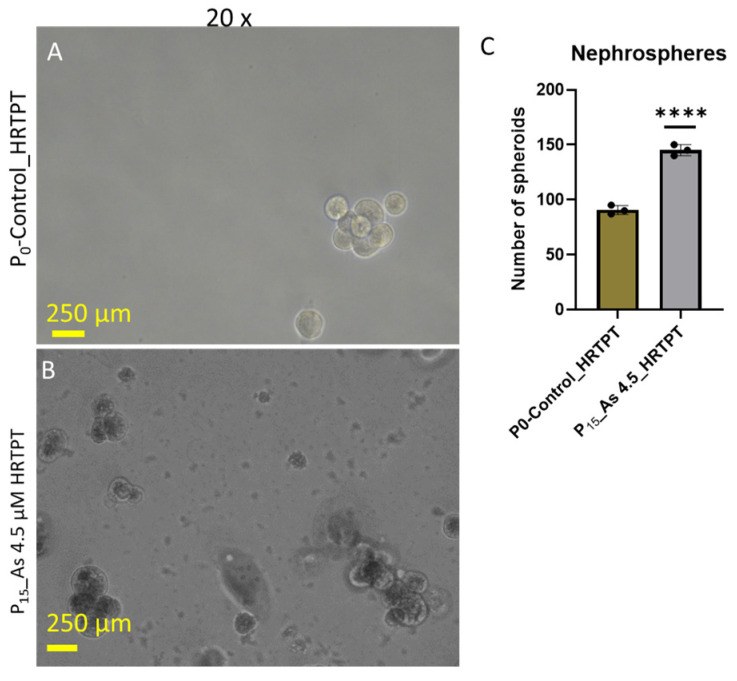
Light-level microscopy of spheroids generated from –Control-HRTPT, i-As (III) 4.5 µM-HRTPT (P_15_) cells. The spheroid images taken at 20× magnification were generated from (**A**) Control-HRTPT cells and (**B**) i-As 4.5 µM HRTPT (P_15_) cells. (**C**) Bar graph shows the number of sphere counts; the measurements were performed in triplicate for control and As (III)-exposed cells. The reported values are mean ± SEM. A *t*-test was performed, and asterisks indicate significant differences from the control (**** *p* < 0.0001). All the images were taken after 21 days of seeding in the ultra-low attachment flasks.

**Figure 5 cells-14-00877-f005:**
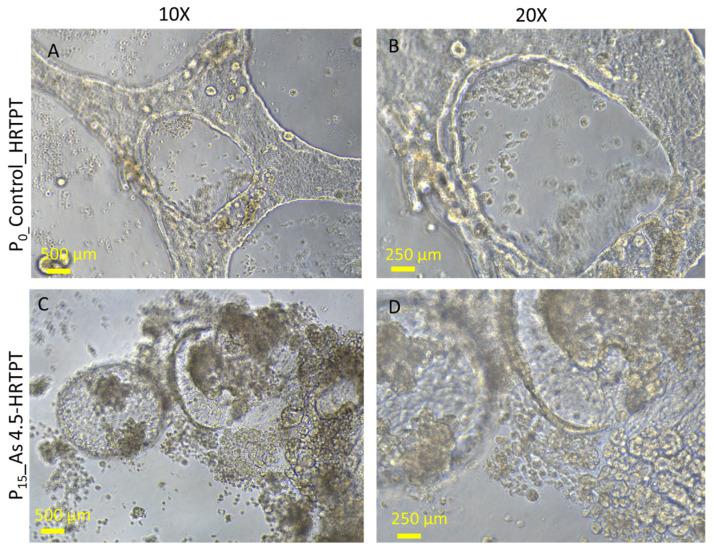
Light-level microscopy of Control-HRTPT and i-As (III) 4.5 µM-HRTPT (P_15_) cells plated on the surface of a thin Matrigel-coated 48-well plate. (**A**,**B**) HRTPT control; (**C**,**D**) As 4.5 µM-HRTPT-P_15_ grown on the surface of the Matrigel coat. All images were taken at 10× and 20× magnification.

**Figure 6 cells-14-00877-f006:**
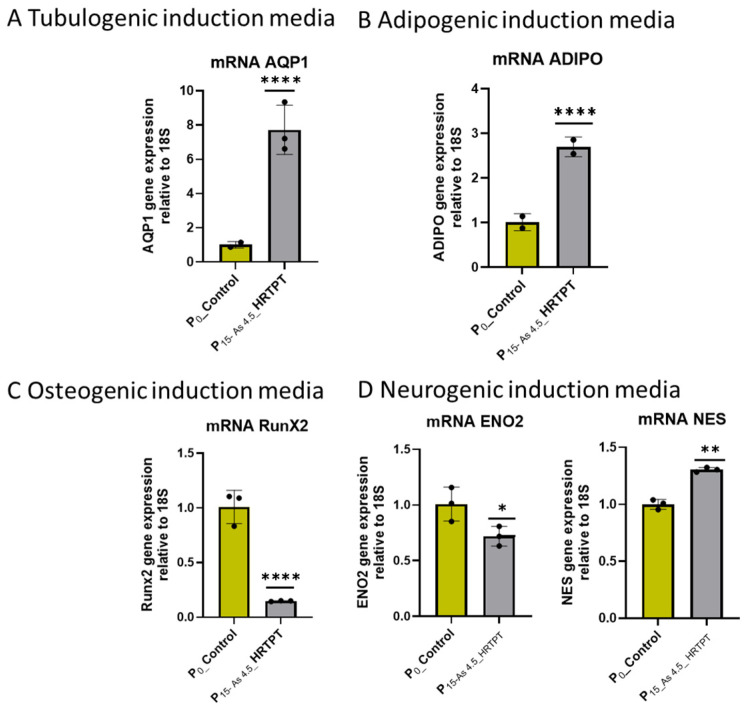
mRNA level of cell-type-specific differentiation markers in Control-HRTPT and P_15_-As 4.5 µM HRTPT. (**A**) mRNA_AQP1 (tubulogenic cell differentiation marker)-RT-qPCR analysis and (**B**) mRNA-ADIPO (adipogenic cell differentiation marker)-RT-qPCR analysis, (**C**) mRNA RunX2 (osteogenic induction marker)-RT-qPCR analysis, (**D**) mRNA_NES and ENO2 (neurogenic cell differentiation marker). Gene expression was normalized to the 18S housekeeping gene. The measurements were performed in triplicate for gene data. The reported values are mean ± SEM. A *t*-test was performed, and asterisks indicate significant differences from the control (* *p* < 0.05, ** *p* < 0.01, **** *p* < 0.0001).

**Figure 7 cells-14-00877-f007:**
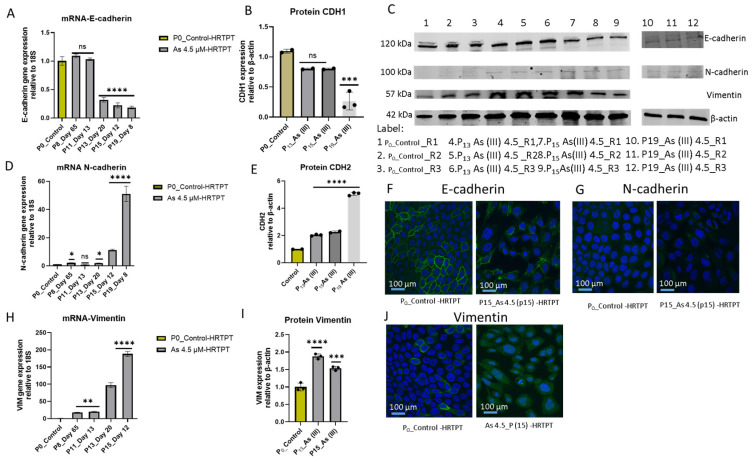
mRNA and protein levels of EMT markers in HRTPT cell lines exposed to 4.5 µM of i-As up to P_19_ passages. (**A**) mRNA_E-cadherin; (**B**) protein_CDH1; (**C**) Western blot results confirmed protein level expression; (**D**) mRNA_N-cadherin; (**E**) protein_CDH2; (**H**) mRNA Vimentin; (**I**) protein vimentin, RT-qPCR and Western blot analysis for protein; ****; ***; **; * indicate significant differences in protein expression level compared to the control 0.0 µM arsenite concentration at *p*-value of ≤0.0001; ≤0.001; ≤0.01; ≤0.05, respectively. Immunofluorescence of CDH1 (**F**), CDH2 (**G**), and Vimentin (**J**) in HRTPT cell line exposed to 4.5 µM of As (III) (P_15_); (**F**) CDH1 expression of control vs. As (III) (P15); (**G**) CDH2 expression of control vs. As^+3^(P_15_); Vimentin expression of control vs. i-As (III) (P_15_); green represents the expression of CDH1/CDH2/Vim, and blue represents DAPI.

**Figure 8 cells-14-00877-f008:**
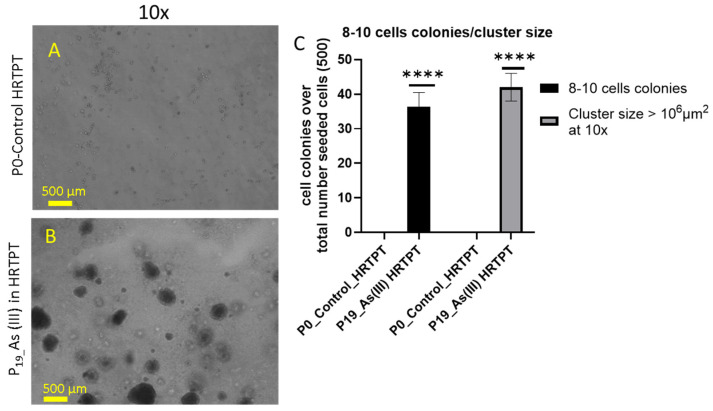
Light-level microscopy of Control-HRTPT and chronic As (III)-exposed cells (P_19_) plated on a soft agar plate up to 28 days. (**A**) HRTPT control; (**B**) chronic i-As (III)-exposed cells grown on soft agar. (**C**) *t*-test for control vs. P_19_ i-As^+3^; 8–10 cell colonies and cluster size > 10^6^ μm^2^. The reported values are mean ± SEM. A *t*-test was performed, and asterisks indicate significant differences from the control (**** *p* < 0.0001). Images were taken at 10× magnification at 28 days after seeding.

**Figure 9 cells-14-00877-f009:**
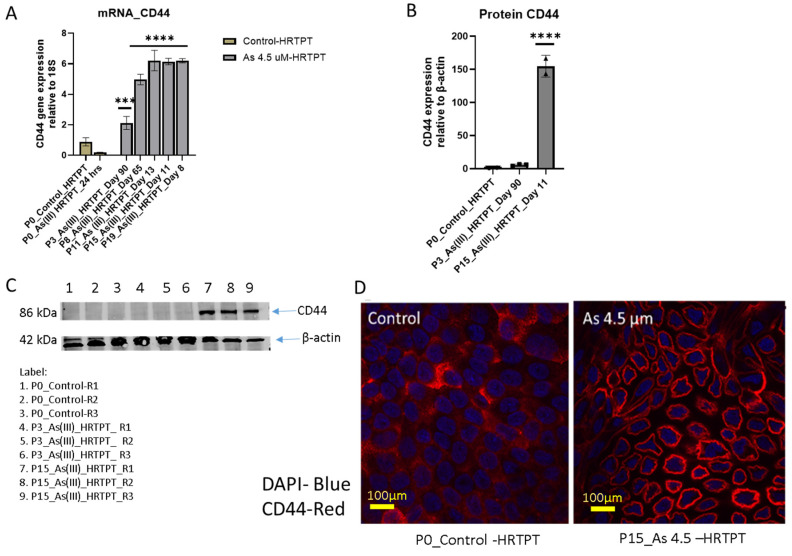
mRNA and protein level of CD44 markers in HRTPT cell lines exposed to 4.5 µM of i-As (III) up to P_19_ passages. (**A**) mRNA_CD44-RT-qPCR analysis; (**B**,**C**) protein_CD44-Western analysis; (**D**) immunofluorescence of CD44 control vs i-As (III)-exposed cells (P_15_); expression of the CD44 gene and the protein was normalized to the 18S housekeeping gene and β-actin, respectively. The measurements were performed in triplicate for gene and protein data. The reported values are mean ± SEM. A one-way Anova was performed, and **** and *** indicate significant differences in gene/protein expression level compared to the control (0.0 µM arsenite concentration at *p*-value of ≤0.0001; ≤0.001, respectively).

**Figure 10 cells-14-00877-f010:**
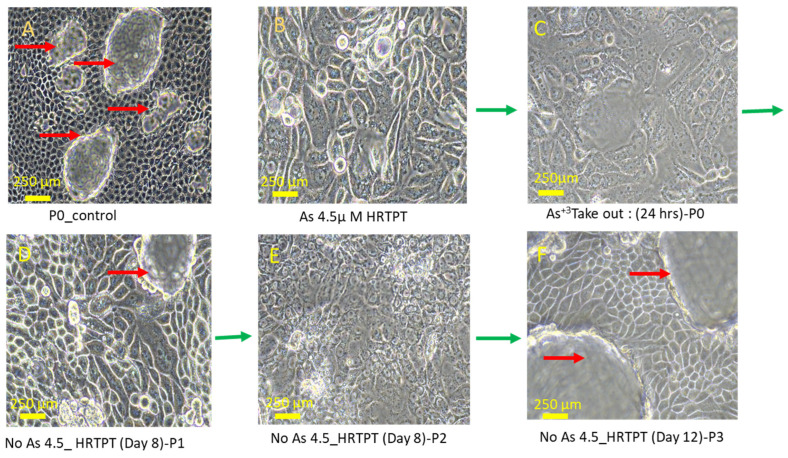
Light microscopic images of As^3+^-removed HRTPT cells from i-As (III)-exposed cells; (**A**) unexposed control; (**B**) i-As (III)-exposed HRTPT cells; (**C**) i-As (III)-removed cells for 24 h (P0); (**D**) P_1_ i As (III)-removed cells; (**E**) P_2_ i-As (III)-removed cells; (**F**) P_3_ i-As (III)-removed cells. The green arrow indicates images of i-As (III)-removed passages; scale bar = 500 µm and magnification 20×. Red arrows show the presence of domes, and green arrows show the chronology of arsenic removal from chronic arsenite-exposed cells (**B**–**F**).

**Figure 11 cells-14-00877-f011:**
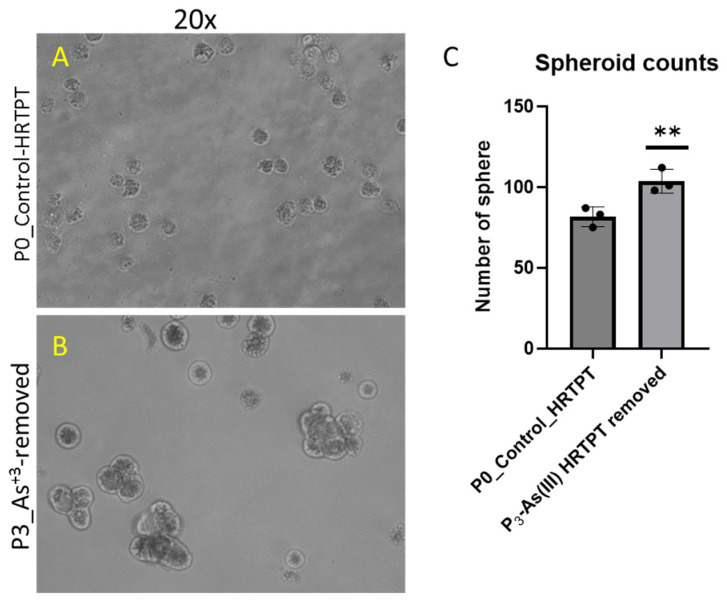
Light-level microscopy of spheroids generated from control-HRTPT and P_3_ i-As (III)-removed-HRTPT cells. The spheroid images taken at 20× magnification generated from (**A**) control-HRTPT cells; (**B**) P_3__i-As (III)-removed-HRTPT cells; and (**C**) bar graph shows number of sphere; ** indicates significant differences in number of spheres compared to the control (0.0 µM arsenite concentration at *p*-value of ≤ 0.01). All the images were taken after 28 days of seeding in the ultra-low attachment flasks.

**Figure 12 cells-14-00877-f012:**
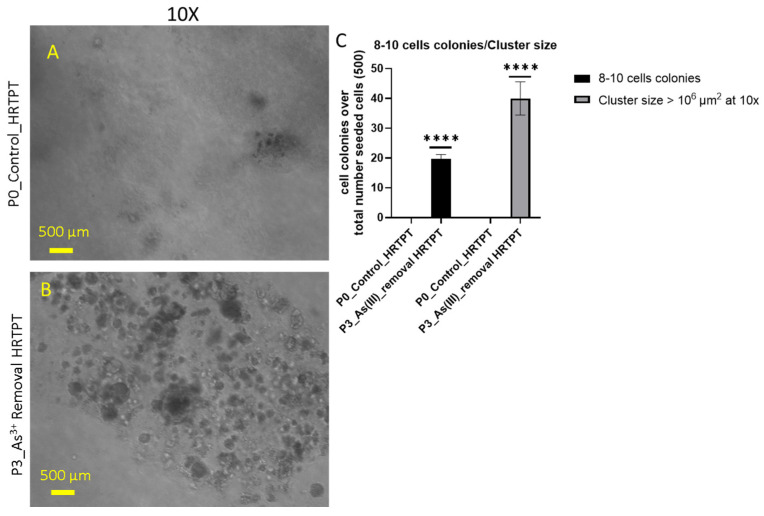
Light-level microscopy of control-HRTPT and P_3__i-As (III)-removed cells plated on a soft agar plate up to 28 days. (**A**) HRTPT control; (**B**) P_3__i-As (III)-removed cells grown on soft agar. (**C**) *t*-test for control vs. As (III)-removed cells; 8–10 cell colonies, and cluster size > 10^6^ μm^2^. The reported values are mean ± SEM. A *t*-test was performed, and asterisks indicate significant differences from the control (**** *p* < 0.0001). Images were taken at 10× magnification at 28 days after seeding. All images were taken at 10× magnification at 28 days after seeding.

**Figure 13 cells-14-00877-f013:**
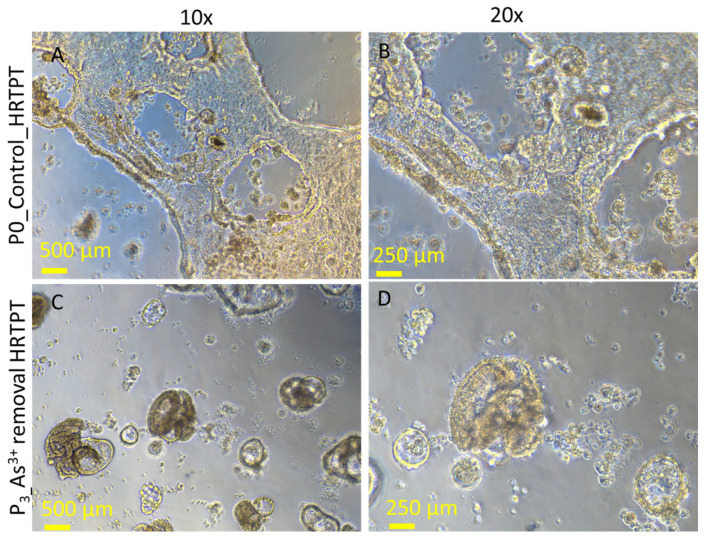
Light-level microscopy of control-HRTPT and P_3_ i-As (III)-removed cells plated on the surface of a thin Matrigel-coated 48-well plate. (**A**,**B**) HRTPT control; (**C**,**D**) P_3_ i-As (III)-removed cells grown on the surface of the Matrigel coat. All images were taken at 10× and 20× magnification.

**Figure 14 cells-14-00877-f014:**
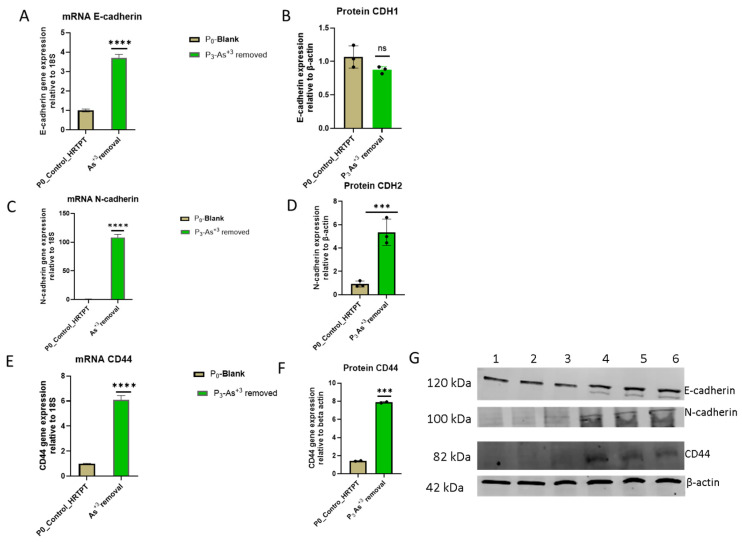
mRNA and protein level of EMT markers in HRTPT cell lines after removal of arsenite vs. non-I-As (III) exposure control. (**A**) mRNA_E-cadherin-RT-qPCR analysis and (**B**,**G**) protein_CDH1–Western blot analysis, (**C**) mRNA_N-cadherin-RT-qPCR analysis (**D**,**G**) protein_CDH2-Western blot analysis, (**E**) mRNA_CD44-RT-qPCR analysis (**F**,**G**) protein_CD44-Western blot analysis. Gene expression was normalized to the 18S housekeeping gene. The measurements were performed in triplicate for gene data. The reported values are mean ± SEM. A *t*-test was performed for mRNA and protein, respectively. Asterisks indicate significant differences from the control (*** *p* < 0.001, **** *p* < 0.0001). “ns” indicates non-significant.

**Figure 15 cells-14-00877-f015:**
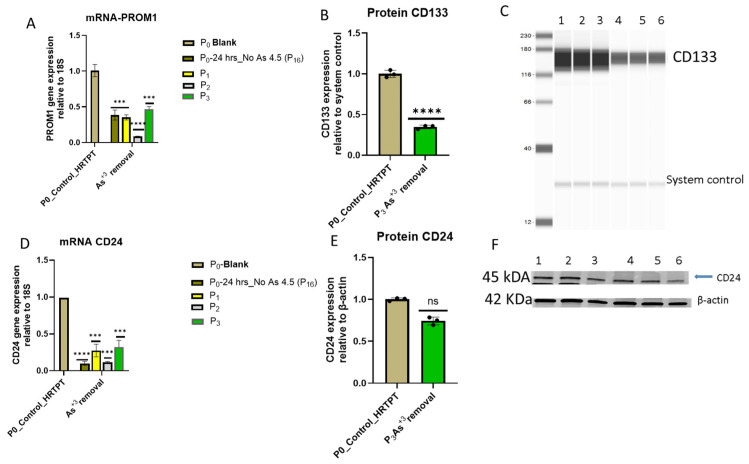
mRNA and protein level of progenitor markers in HRTPT cell lines after removal of arsenite vs. non-arsenite exposure. (**A**) mRNA_CD133- RT-qPCR analysis and (**B**,**C**) protein_CD133 -Jess analysis; (**D**) mRNA_CD24- RT-qPCR analysis; (**E**,**F**) protein_CD24-Western blot analysis; gene expression was normalized to the 18S housekeeping gene. The measurements were performed in triplicate for gene data. The reported values are mean ± SEM. A one-way ANOVA and *t*-test were performed for mRNA and protein, respectively. Asterisks indicate significant differences from the control (*** *p* < 0.001, **** *p* < 0.0001). “ns” indicates non-significant.

**Table 1 cells-14-00877-t001:** Shows the extended time course of i-As (III) exposure.

Number of Passages	Days to Next Passage at 4.5 μM of As^+3^
P_0_	1
P_1_	7
P_2_	25
P_3_	90
P_4_	13
P_5_	18
P_6_	45
P_7_	40
P_8_	65
P_9_	37
P_10_	6
P_11_	13
P_12_	12
P_13_	20
P_14_	12
P_15_	11
P_16_	12
P_17_	11
P_18_	7
P_19_	8

“P” indicates passage; control cells passed to P19 and confluence in seven days.

## Data Availability

The original contributions presented in this study are included in the article. Further inquiries can be directed to the corresponding author.

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
