# Peer review of "Inorganic Arsenite [As (III)] Represses Human Renal Progenitor Cell Characteristics and Induces Neoplastic-like Transformation"

_cells, 2025, doi:10.3390/cells14120877_

Round 1
Reviewer 1 Report
Comments and Suggestions for Authors
Haque at al. examine the influence of arsenic on renal progenitor cells with regard to neoplastic transformation. The study and the results are very interesting, but I have a few comments. My main point is that there is a lack of clear information on the number of passages of the control cells.
Introduction:
In the first sentence of the introduction, examples of diseases should perhaps be mentioned.
It is a bit unusual to read that the bladder is near the kidney and that the kidney is therefore exposed to the same concentrations of arsenic. After all, the kidney forms the urine that ends up in the bladder, so perhaps it would be better to refer to the kidney's excretory function rather than its proximity?
I would most likely rephrase the goals. It was already known when the publication was written whether the first goal had been achieved.
Material and Methods:
The explanation of the statistical methods used is missing in material and methods.
Results:
In Figure 1A, the arrows and circles are missing an explanation. Further, scale bars are missing in all microscopy pictures. In 1A, 5A you mention a scale bar, but I do not see it. In 2A, 2B, 4A, 5C, 5D it is not mentioned, in 3, 4B, 5B there is not even a magnification given
Furthermore, it is not clear what the day should mean in addition to the passage. I assume that this is day 65 in passage 8 or day 11 in passage 15, but for that to be the case, it would also have to be written somewhere how long such a passage actually lasts.
Why are there no structures marked with arrows or circles in Figures B and F?
It is not clear from Figure 1B whether the calculation in (B) was based on the western blot or the Jess. It looks like the western blot, but it is not stated.
Why Figure 1A with A to F, Figure 1B with A to D and Figure 1C with A to C? Is this to comply with the number of illustrations? I do not remember having seen this numbering before.
What does 20.25 mean in heading 3.2.?
In section 3.2 or earlier in the introduction, when they are introduced, an explanation is missing as to why renal progenitor cells should differentiate into cells other than tubulogenic ones at all.
It is not clear from Figure 3 and the description of Figure 3 or Material and Methods, whether the control was always passage 0 or also passage 19, under A the only place where the passage for the control is given, it is passage 0. The control should represent cells from the highest passage used for As treatment, since ageing can also change marker expression. The passage number should also be added to the control-HRTPT in all figures.
Figure 4A: only **** exist in this graph, no need to explain all asterisks (please check all legends for this, in Figure 3,… also too many asterisks combinations are explained).
In Figure 5A, I am not sure what the green arrows indicate. After all, the way the cells were treated is given below the images. I am missing an image of the control cells (which are mentioned in the accompanying text) and markers for the various structures visible.
In my opinion, the statements about Figure 5 also suffer from the fact that it is not mentioned which passage the control cells had.
The labeling of the bands of the western blot is missing in Figure 5F.
The original western blots shown are confusing without a label indicating which lanes and bands were used in which figure and which band represents which protein. There are several blots that are the same in Figure 3 and Figure 5F, although if I understood it correctly, in Figure 3 there was arsenic on the cells the whole time and in Figure 5 it was removed. How can this be the same blots then? Furthermore, the blots lack the kD labeling of the marker bands.
Discussion:
The statements in the last third of the first paragraph of the discussion have no reference to literature. For example, where has it been shown that the HRTPT cells exposed to arsenic did not form tumors in immunodeficient mice? Where can the reference be found that 60 doublings of normal cells represent an accepted lifespan?
A difference was found in this study compared to a previous study of the same group in which the removal of arsenic allowed the cells to return to the state of the parent cells, which did not happen here. I am missing an explanation as to why these two studies yielded different results.
Comments on the Quality of English LanguageThere are mainly mistakes in the legends, mainly concerning the use of capital letters where I would not expect them, this is also the case in the rest of the text, e.g. human.
Author Response
Manuscript ID: cells-3578497
Inorganic arsenite [As (III)] represses human renal progenitor cell characteristics and induces neoplastic-like transformation
Md Ehsanul Haque , Swojani Shrestha , Donald Sens , Scott Garrett
Response to Authors’ Comments
Reviewer 1
Reviewer Comment: “My main point is that there is a lack of clear information on the number of passages of the control cells.”
Authors’ Response: Control cells have been a common point of contention, and one school of thought is that a passage-match control should be used. Indeed, we have designed our experiments with these controls and have them if the reviewers maintain their requirements to include them. Arsenite-exposed cells go through extensive stress and crisis conditions, causing a near complete pause in mitosis, most probably a lock in the cell cycle, leading to very large extended times to complete a passage. This aspect of the study was not highlighted in the previous submission, and in this revision, we included a table listing the times to reach confluence (see below and in Results). A proper control sample would be a harmless block in the cell cycle for an equivalent amount of time. We do not think such a control exists for this. Thus, our approach was to simply use the cell culture immediately before exposure.
Reviewer Comment: “In the first sentence of the introduction, examples of diseases should perhaps be mentioned”.
Authors’ Response: We agree that elaborating on this would be helpful. There are no arsenic-specific diseases except for overt arsenic poisoning, which sometimes occurs. Lower-level, environmental exposures have been shown to accelerate and/or enhance other known chronic disease processes. We added the following statement in the Introduction:
Manuscript alteration: “Inorganic arsenic is a toxic substance that is distributed throughout the environment and has been implicated in numerous disease states, such as causing overt organ toxicity, especially in the liver and kidney, and exacerbating cardiovascular disease, neurological deficits and peripheral neuropathy, diabetes, and causing bladder, skin, lung and other cancers.”
Reviewer Comment: “It is a bit unusual to read that the bladder is near the kidney and that the kidney is therefore exposed to the same concentrations of arsenic. After all, the kidney forms the urine that ends up in the bladder, so perhaps it would be better to refer to the kidney's excretory function rather than its proximity?”
Authors’ Response: We are attempting to convey that it is likely that the kidney is routinely exposed to similar concentrations of arsenic as those of the bladder, and, indeed, this is really due to the function rather than anatomic proximity.
Manuscript alteration: “Human bladder cancer is strongly associated with exposure to i-As (III) present in drinking water and since the kidney is proximal to the bladder intimately involved in the excretion of the metalloid upstream to the bladder, the cellular components of the kidney are exposed to at least the same level of i-As (III) as the bladder(10)”.
Reviewer Comment: “I would most likely rephrase the goals. It was already known when the publication was written whether the first goal had been achieved.”
Authors’ Response: Technically, the reviewer is right in that our previous study (reference 20), we observed a considerable decrease in Prom1 and CD24 at the protein level. Thus, we rephrased the goal as follows:
Manuscript alteration: “The goal of the study was to assess whether the cells would maintain or increase the repression of Prom1 and CD24 upon extended i-As(III) exposure and whether this could result in the conversion to the loss of progenitor properties of the HRTPT cells with progression to those displaying characteristics of malignant transformation.”
Reviewer Comment: Material and Methods: The explanation of the statistical methods used is missing in the Materials and Methods.
Authors’ Response: Indeed, it was, and an oversight on our part. We included a Statistical Analysis section under section 2.11
Manuscript alteration: The experiments were carried out in triplicate, and the data were analyzed using one-way ANOVA (non-parametric) with Tukey post-hoc testing using GraphPad PRISM 10. Gene expression was normalized to the 18S ribosomal gene. The measurements were performed in triplicate for gene data. The reported values are mean ± SEM. A t-test was performed for mRNA and protein, respectively. And asterisks indicate significant differences from the control (*p< 0.05,** p < 0.01, *** p < 0.001, **** p < 0.0001).
Reviewer Comment: In Figure 1A, the arrows and circles are missing an explanation. Further, scale bars are missing in all microscopy pictures. In 1A, 5A you mention a scale bar, but I do not see it. In 2A, 2B, 4A, 5C, 5D it is not mentioned, in 3, 4B, 5B there is not even a magnification given.
Authors’ Response: Indeed, these items are missing. Scale bars were added to all micrographs.
Manuscript alteration: For Figure 1, we added, “Red arrows show dome-like structures, Blue arrows show fibroblast-like structures, and Red circles show myofibroblast-like cells.” For Figure 10 we added: “Red arrows show the presence of domes, and Green arrows designate after arsenite removal from chronic arsenite-exposed cells (C-F).”
Reviewer Comment: Furthermore, it is not clear what the day should mean in addition to the passage. I assume that this is day 65 in passage 8 or day 11 in passage 15, but for that to be the case, it would also have to be written somewhere how long such a passage actually lasts.
Authors’ Response: We realize that there is some confusion with the passage number and the time it took for cells to reach confluence at each passage. We added a table (Table I) that depicts the increased time for cells to reach confluence in the early passages and the ability of the cells to grow faster in the continued presence of the metalloid. This is an important concept in the paper that we failed to explicitly bring out.
Table 1. The number of days for HRTPT cells to reach confluence in the presence of 4.5 μM iAs(III) at each passage number.
Number of Passages |
Days to Next Passage at 4.5 μM As+3 |
P0 |
1 |
P1 |
7 |
P2 |
25 |
P3 |
90 |
P4 |
13 |
P5 |
18 |
P6 |
45 |
P7 |
40 |
P8 |
65 |
P9 |
37 |
P10 |
6 |
P11 |
13 |
P12 |
12 |
P13 |
20 |
P14 |
12 |
P15 |
11 |
P16 |
12 |
P17 |
11 |
P18 |
7 |
P19 |
8 |
Manuscript alteration: The following was added as the second sentence under heading 3.1: “Table 1 shows that during the initial passages (P1 to P9), the cells struggled to reach confluence in the continued presence of the metalloid, with five of the passages taking more than one month to fill the flask, and P3 taking three months. In later passages (P10 to P19), the cells were able to grow and divide during continued exposure to 4.5 µM i-As(III) at a much greater rate, taking only between one and two weeks per passage.”
Reviewer Comment: Why are there no structures marked with arrows or circles in Figures B and F?
Authors’ Response: Micrographs B and F now have red arrows depicting dome structures.
Reviewer Comment: It is not clear from Figure 1B whether the calculation in (B) was based on the western blot or the Jess. It looks like the western blot, but it is not stated
Authors’ Response: Our lead author performed both western blot and capillary electrophoresis with the Jess instrument. We have just begun implementing this new instrument in our laboratory, and thus, the current manuscript has used both methods. For the quantification of fold induction of protein levels, both methods were used, and in each case, the non-exposed controls were used to assess fold induction.
Manuscript alteration: Figure 2. B) Protein CD133 analysis (quantification from images in C and D). (C) Westen blot (D) Capillary Western/Jess of CD133
Reviewer Comment: Why Figure 1A with A to F, Figure 1B with A to D and Figure 1C with A to C? Is this to comply with the number of illustrations? I do not remember having seen this numbering before.
Authors’ Response: We agree that the numbering system is confusing. We have renumbered the figures.
Manuscript alteration: Fig. 1 A changed to Fig 1, Figure 1B changed to Fig 2, and Figure 1C changed to Fig. 3.
Reviewer Comment: What does 20.25 mean in heading 3.2.?
Authors’ Response: This is simply a mistake that we failed to correct.
Manuscript alteration: 20.25 has been deleted in heading 3.2
Reviewer Comment: In section 3.2 or earlier in the introduction, when they are introduced, an explanation is missing as to why renal progenitor cells should differentiate into cells other than tubulogenic ones at all.
Authors’ Response: This is a fair question and one that provokes curiosity. It is not uncommon for stem cells to have propensities to differentiate into other cell types if given the right signals or environments. In our original publications on these cells, we tested their abilities to differentiate into other cell types. In actual fact, CD133+/CD24+ cells taken directly from renal cortex have been shown to exhibit adipocyte, neuroblast, and osteocyte differentiation in an old 2006 publication, and this is the origin of these assays. The citation for this study was placed in the manuscript in the paragraph describing the transdifferentiation results.
Manuscript alteration: In section 3.2: The ability of the i-As (III) exposed cells at P15 to undergo differentiation was tested using gene expression markers commonly used to provide evidence for tubulogenic, adipogenic, osteogenic, and neurogenic differentiation in the HRTPT cells 29, 46. CD133+/CD24+ cells, isolated from human kidney, have been demonstrated to transdifferentiate into these cell types (ref—Appel et al., 22006). The results showed that the markers for tubulogenic differentiation (AQP1) and adipogenic differentiation (ADIPO) were significantly elevated and the marker for osteogenic differentiation (RUNX2) significantly decreased compared to control cells (Figure 6).
Appel D et. al., Recruitment of Podocytes from Glomerular Parietal Epithelial Cells. J Am Soc Nephrol 2009k, 20(20): 333-343.
Reviewer Comment: It is not clear from Figure 3 and the description of Figure 3 or Material and Methods, whether the control was always passage 0 or also passage 19, under A the only place where the passage for the control is given, it is passage 0. The control should represent cells from the highest passage used for As treatment, since ageing can also change marker expression. The passage number should also be added to the control-HRTPT in all figures.
Authors’ Response: As was mentioned in the first comment, we used the passage immediately before exposure for the control. Passage-matched controls, we felt, were not proper controls, since the exposed cells took several weeks to several months to reach confluency during arsenite exposure. Without arsenite exposure, the cultures reach confluency in one week, whereas with the exposures, they take several weeks, and passage 8 took 65 days, as is now shown in Table 1. It is difficult to have a control sample to match for this. These cells struggle to gain confluency over a couple of months. Simply having the cells harvested with the equivalent number of doublings is not really a direct control. Cells seldom change in several passages, and the overwhelming issue with arsenic exposure was to overcome the toxicity to fill the flask. The passage-match for that culture is really quite divergent from the experimental group. We think the earlier passages with arsenite exposure are the true control because these cells have gone through the struggle of reaching confluency with arsenite but have yet to undergo the molecular changes in the various genes we are measuring. We do, however, have passage-matched control samples saved and were packaged into the overall experimental design.
Reviewer Comment: Figure 4A: only **** exist in this graph, no need to explain all asterisks (please check all legends for this, in Figure 3,… also too many asterisks combinations are explained).
Authors’ Response: We agree, and we have deleted those symbols that are not depicted in the figure.
Reviewer Comment: In Figure 5A, I am not sure what the green arrows indicate. After all, the way the cells were treated is given below the images. I am missing an image of the control cells (which are mentioned in the accompanying text) and markers for the various structures visible.
Authors’ Response: We added a micrograph of the control cells, and we explained the arrows.
Manuscript alteration: Red arrow shows the presence of domes, and Green arrows show the strategy of arsenic removal from chronic arsenite-exposed cells (B-F).
Reviewer Comment: In my opinion, the statements about Figure 5 also suffer from the fact that it is not mentioned which passage the control cells had.
Authors’ Response: All controls are unexposed cells at the initial passage.
Reviewer Comment: The labeling of the bands of the western blot is missing in Figure 5F.
Authors’ Response: The bands in old Figure 5F, now Figure 15 are labeled.
Reviewer Comment: The original western blots shown are confusing without a label indicating which lanes and bands were used in which figure and which band represents which protein. There are several blots that are the same in Figure 3 and Figure 5F, although if I understood it correctly, in Figure 3 there was arsenic on the cells the whole time and in Figure 5 it was removed. How can this be the same blots then? Furthermore, the blots lack the kD labeling of the marker bands.
Authors’ Response: This is an embarrassing oversight on our part. We included a new original Western blot image file, lanes labelled in the original file, and also labelled in Old Fig.3 or New Fig.7.
Reviewer Comment: The statements in the last third of the first paragraph of the discussion have no reference to literature. For example, where has it been shown that the HRTPT cells exposed to arsenic did not form tumors in immunodeficient mice? Where can the reference be found that 60 doublings of normal cells represent an accepted lifespan?
Authors’ Response: We thought that we had published this negative finding in the previous publication, where we had exposed these cells to arsenite. We performed these experiments on cells that were at passage 10, and we are officially entering this observation in the current publication. We added a section in the Materials and Methods for tumors in Nude Mice.
As for the reference that 60 doublings of normal cells represent an accepted lifespan, it relates to the original publication that is commonly referenced from Hayflick, 1965. We added that reference to the statement.
Manuscript alteration:
2.11 Tumor formation in immunocompromised mice
“The ability of the iAs-exposed cells to form tumors was tested by the injection of 1 x 106 cells at passage 16 in the dorsal thoracic midline of athymic nude mice (NCRnu/nu) as described previously (Sandquist et al., 2016). The mice were observed every two weeks for 12 weeks for any sign of tumor formation (raised area at the injection site. Following euthanasia, the injection site of all 5 mice was dissected, and the area was observed for evidence of tumor growth. This study was approved by the UND IACUC#1911- 1C.”
Results section 3.4
“An attempt to assess tumor formation in vivo using cells passaged in 4.5 μM i-As(III) from our previous study (reference 31). The examination of the 5 mice injected with iAs-exposed cells at p16 showed no visible evidence of tumor formation or following examination of the tissue at the injection site.”
References
Hayflick L. Exp. Cell Res. 1965, The limited in vitro lifetime of human diploid cell strains.
THE LIMITED IN VITRO LIFETIME OF HUMAN DIPLOID CELL STRAINS.
HAYFLICK L. Exp Cell Res. 1965 Mar;37:614-36. doi: 10.1016/0014-4827(65)90211-9. PMID: 14315085
PLoS One 2016 May 25;11(5):e0156310. doi: 10.1371/journal.pone.0156310. eCollection 2016.
Loss of N-Cadherin Expression in Tumor Transplants Produced From As+3- and Cd+2-Transformed Human Urothelial (UROtsa) Cell Lines
Elizabeth J Sandquist 1 , Seema Somji 1 , Jane R Dunlevy 2 , Scott H Garrett 1 , Xu Dong Zhou 1 , Andrea Slusser-Nore 1 , Donald A Sens 1
Reviewer Comment: A difference was found in this study compared to a previous study of the same group in which the removal of arsenic allowed the cells to return to the state of the parent cells, which did not happen here. I am missing an explanation as to why these two studies yielded different results.
Authors’ Response: The current study extended the culture of HRTPT from passage 10 in the previous study to passage 19 in the current study. The different result is due to the extension of toxicity and the adaptation of the cells to increase their growth in the presence of arsenic.
Manuscript alteration: No alteration to the manuscript

Reviewer 2 Report
Comments and Suggestions for Authors
- PROM 1 and CD24 biomarkers of Repair and regeneration of damaged tubular epithelium? What’s the CD44? It needs to explain, not just abstract.
- The organization and presentation of the figures is a bit messy and needs to be reformatted. Fig 1A-A, Fig 1A-B? Make sure that the number of the figures is the same as the number quoted in the body,
Author Response
Reviewer 2
Reviewer Comment: PROM 1 and CD24 biomarkers of Repair and regeneration of damaged tubular epithelium? What’s the CD44? It needs to explain, not just in the abstract.
Authors’ Response: During extensive damage to the tubular epithelium, those cells that regenerate the lost epithelial cells possess PROM1 and CD24. As to the induction of CD44 and its ramifications, we added a large paragraph in the discussion about this, starting from its overall function, to its role in fibrosis and in tumor progression.
Manuscript alteration: Inserted before the last paragraph of the Discussion.
CD44 fared prominently in the current study, having roles in long-term responses to injury and fibrosis and being a prognostic marker for various cancers, including ccRCC. CD44 is a cell surface transmembrane glycoprotein that functions in cell adhesion, migration, and lymphocyte homing. It is expressed on a variety of cell types, including granulocytes, lymphocytes, fibroblasts, and even in epithelial cells. Functional ligands include hyaluronic acid, osteopontin, collagens, and matrix metalloproteinases, and CD44 activation initiates a variety of cell signaling pathways through interactions with intracellular kinases. Many of these activities and pathways are also commonly exploited by tumor cells to influence tumor growth, metastasis, and resistance to therapy. Thus, CD44 is a common marker of various types of cancers, including renal cell carcinoma. Important for the current study is the role of CD44 in the process of fibrosis. Since CD44 has many interactions with the extracellular matrix, there are many potential ways this surface protein can facilitate processes in fibrosis, such as the recruitment of fibroblasts, promoting the deposition of matrix, and also the recruitment of inflammatory cells (the two references). Recently, Matsushita et al 2024 showed that CD44 promoted renal fibrosis by secreting fibronectin from renal tubular epithelial cells (TEC) during partial epithelial-mesenchymal transition in response to renal injury. CD44 was expressed in atrophic, dilated, and hypertrophic TECs with fibrotic lesions during chronic injury. By laser microdissection, these TECs were collected and analyzed using microarrays. Gene ontology analysis suggested that these TECs had a mesenchymal phenotype, and pathway analysis identified CD44 as an upstream regulator of fibrosis-related genes, including fibronectin 1 (Fn1). This study shows that CD44 expression marks renal tubular epithelial cells undergoing maladaptive repair and promotes the development of renal fibrosis18. For roles in cancer, most solid malignancies contain cancer stem cells characterized by CD44. As a result of hyaluronic acid interaction with CD44, EGFR-mediated pathways are activated, leading to tumor cell growth, tumor cell migration, and chemotherapy resistance in solid cancers19.
Other research observed that an active CD44/p-AKT/AKT signaling pathway promotes resistance to FGFR1 inhibition in squamous-cell lung cancer20. An in-depth study of the roles of CD44/CD24 and ALDH1 as cancer stem cell markers in tumorigenesis and metastasis21, 22. The CD44 gene is a major molecular prognostic marker for cancer stem cells (CSCs) of many solid malignancies, including RCC, as well as one of the major components of surviving tubular cells in renal fibrosis 21, 23-25. The expression of CD44 in clear cell renal cell carcinoma (ccRCC) correlates with tumor grade and patient survival, influenced by the methylation of genes 26. Furthermore, higher CD44 expression is associated with more aggressive behavior, tumor progression, and a worse prognosis for RCC 21, 27, 28. The current study shows that CD44 was induced in the late phases of the exposure timeline and appears to have a dual role in promoting maladaptive and fibrotic-like repair as well as promoting transformation. It is important to note that, according to Figure 9D, CD44 was expressed in nearly all epithelial cells, but in Figure 12, only 5 to 10% of the cells were able to form colonies in soft agar. Thus, CD44 is likely to function in promoting fibrosis during chronic injury for the majority of cells, and for those cells that have undergone transformation, to potentially enhance their tumorigenicity and/or progression.
Reviewer Comment: The organization and presentation of the figures is a bit messy and needs to be reformatted. Fig 1A-A, Fig 1A-B? Make sure that the number of the figures is the same as the number quoted in the body,
Authors’ Response: We agree, and this was a comment from another reviewer. We have overhauled the figure numbering system with figures 1 through 15
Manuscript alteration: Figures renumbered 1 through 15
Reviewer 3 Report
Comments and Suggestions for Authors
Comments: The authors present interesting data on continuous 4.5uM arsenite exposure to proximal tubule progenitor cells (HRTPT) that shows reduction in CD24 and PROM1 surface markers but induction of CD44, a common surface marker cancer stem cells that promotes epithelial-mesenchymal transition (EMT). The work is well written and referenced; it is a nice extension of their recent arsenite work with HRTPT cells (Singhal et al 2023) progressing toward an EMT phenotype. HRTPT were sorted for CD133+/CD24+ markers from RPTEC/TERT1 immortalized cells along with CD133-/CD24+ cells (HREC24T). Each of these two cell subpopulations have different phenotypes from the original RPTEC/TERT1 cell line, and with arsenite exposure these differences in phenotype and surface markers continue.
My major comment is that continual arsenite exposure constitutes a selection pressure in culture that can promote overgrowth of a subpopulation presenting with an EMT phenotype and surface markers. One could argue HRTPT pleiotropy accounts for phenotypic adaptation to arsenite exposure, but after P25, it is unlikely that cells will revert to their original marker levels and phenotype, and transcriptome. The authors do mention progenitor cell heterogeneity in the final paragraph of the Discussion, but a more complete discussion of selection pressure would be worthwhile and as the author imply, scRNA-seq, to lend further insight into this question of subpopulation outgrowth.
Minor points: On pg 9, ln 312, please change to “…starting at P12…”. On pg17, ln 510, please remove the hyphen on “enlarge”.
Author Response
Reviewer 3
Reviewer Comment: My major comment is that continual arsenite exposure constitutes a selection pressure in culture that can promote overgrowth of a subpopulation presenting with an EMT phenotype and surface markers. One could argue HRTPT pleiotropy accounts for phenotypic adaptation to arsenite exposure, but after P25, it is unlikely that cells will revert to their original marker levels and phenotype, and transcriptome. The authors do mention progenitor cell heterogeneity in the final paragraph of the Discussion, but a more complete discussion of selection pressure would be worthwhile, and as the author imply, scRNA-seq, to lend further insight into this question of subpopulation outgrowth.
Authors’ Response: See Manuscript alteration.
Manuscript alteration: Cell culture and cell population dynamics likely play a role in the adaptation process. The original telomerase-immortalization process for the original cell line was performed in a culture of renal epithelial cells established from a human renal cortex sample, and once established, it did not undergo cloning (Hayflick, 1965). Since the HRTPT CD133+/CD34+ cell line was derived from this, the cell population of the experimental culture is likely heterogeneous. In explaining the adaptation to long-term i-As(III), it is possible that various subtypes of epithelial cells gradually predominate in the culture after 19 passages. Several observations argue against this explanation. First, in our previous study (20, Singhal, 2023), cells exposed up to passage 9 and then recovered in normal media that lacked the metalloid, nearly totally reverted their mesenchymal-like gene expression back to that of the global gene expression profile of the original epithelial culture. After passage 19, that did not occur. Another observation that suggests molecular changes and not selection of a subpopulation is the predominant explanation for adaptation to i-As(III) is the sudden and near-complete expression of CD44. This protein was expressed in nearly every cell once induced, as evidenced by fluorescent staining. A gradual selection process would likely result in CD44 expression in a fraction of the total population. Since CD44 has been observed in chronically injured and fibrosed kidneys (ref), gives further likelihood that this is a real molecular gene expression change from i-As(III) cell stress and injury.
Reviewer Comment: Minor points: On pg 9, ln 312, please change to “…starting at P12…”. On pg17, ln 510, please remove the hyphen on “enlarge”.
Authors’ Response: We have made these appropriate changes to the manuscript.
Round 2
Reviewer 1 Report
Comments and Suggestions for Authors
Dear Authors,
although some of my comments have been answered satisfactorily, but my main concern remains.
Reviewer Comment: In Figure 1A, the arrows and circles are missing an explanation. Further, scale bars are missing in all microscopy pictures. In 1A, 5A you mention a scale bar, but I do not see it. In 2A, 2B, 4A, 5C, 5D it is not mentioned, in 3, 4B, 5B there is not even a magnification given.
Authors’ Response: Indeed, these items are missing. Scale bars were added to all micrographs.
Reviewer Comment Revision: Figs 4B, 7F, G, J, 9D are still missing the scale bar.
Reviewer Comment Revision: Sorry, maybe I missed it, but how long is a passage for control cells, since it is now stated under Table 1, that P15 and P19 were similar to control cells.
How many times have you done the passaging to gather these data fort he days to next passage?
Reviewer Comment: It is not clear from Figure 3 and the description of Figure 3 or Material and Methods, whether the control was always passage 0 or also passage 19, under A the only place where the passage for the control is given, it is passage 0. The control should represent cells from the highest passage used for As treatment, since ageing can also change marker expression. The passage number should also be added to the control-HRTPT in all figures.
Authors’ Response: As was mentioned in the first comment, we used the passage immediately before exposure for the control. Passage-matched controls, we felt, were not proper controls, since the exposed cells took several weeks to several months to reach confluency during arsenite exposure. Without arsenite exposure, the cultures reach confluency in one week, whereas with the exposures, they take several weeks, and passage 8 took 65 days, as is now shown in Table 1. It is difficult to have a control sample to match for this. These cells struggle to gain confluency over a couple of months. Simply having the cells harvested with the equivalent number of doublings is not really a direct control. Cells seldom change in several passages, and the overwhelming issue with arsenic exposure was to overcome the toxicity to fill the flask. The passage-match for that culture is really quite divergent from the experimental group. We think the earlier passages with arsenite exposure are the true control because these cells have gone through the struggle of reaching confluency with arsenite but have yet to undergo the molecular changes in the various genes we are measuring. We do, however, have passage-matched control samples saved and were packaged into the overall experimental design.
Reviewer Comment Revision: Sorry, I am still not convinced that P0 is sufficient as control. I understand that it is not easy to find a suitable control, but at least non-exposed cells in P19 should be included, especially if you already have the data. Further, this problem should be extensively discussed.
I completely disagree with your statement that cells do not change within several passages, if I understand it right. We use various kidney cells, some of which change within 10 to 15 passages without toxin exposure and can no longer be used due to the loss of important receptors. I am quite relieved that your reference to the acceptable 60 passages is so old, as it cannot cover all existing new cultured cells.
Reviewer Comment: Figure 4A: only **** exist in this graph, no need to explain all asterisks (please check all legends for this, in Figure 3,… also too many asterisks combinations are explained).
Authors’ Response: We agree, and we have deleted those symbols that are not depicted in the figure.
Reviewer Comment Revision: Please also do this in Fig 2, 3, 6 (***), 7, 9, 11, 12, 14, 15. It was a bit tedious having to check everything again.
Reviewer Comment: In Figure 5A, I am not sure what the green arrows indicate. After all, the way the cells were treated is given below the images. I am missing an image of the control cells (which are mentioned in the accompanying text) and markers for the various structures visible.
Authors’ Response: We added a micrograph of the control cells, and we explained the arrows.
Reviewer Comment Revision: If, as mentioned in the text, the cells from which As was removed displayed no domes, why are there red arrows pointing to domes in the pictures?
Reviewer Comment: The original western blots shown are confusing without a label indicating which lanes and bands were used in which figure and which band represents which protein. There are several blots that are the same in Figure 3 and Figure 5F, although if I understood it correctly, in Figure 3 there was arsenic on the cells the whole time and in Figure 5 it was removed. How can this be the same blots then? Furthermore, the blots lack the kD labeling of the marker bands.
Authors’ Response: This is an embarrassing oversight on our part. We included a new original Western blot image file, lanes labelled in the original file, and also labelled in Old Fig.3 or New Fig.7.
Reviewer Comment Revision: Well, now you have included Figure numbers (but they seem to fit to the old Figure numbers) in the file oft he original blots, but still there are questions:
Fig 1B is only the graph, CD24 is not shown in Fig 1C, CD133 does not look the same in 1C as in the supposed original. Where is the b-actin blot corresponding to 1C.
Why was the intensity of the Jess in 1D changed compared to the original but not stated?
And here, I stopped since I do not have the time to check all the blots. You should do so again.
Author Response
Word file enclosed.
